# Polysaccharide from Okra (*Abelmoschus esculentus* (L.) Moench) Improves Antioxidant Capacity via PI3K/AKT Pathways and Nrf2 Translocation in a Type 2 Diabetes Model

**DOI:** 10.3390/molecules24101906

**Published:** 2019-05-17

**Authors:** Zhengzheng Liao, Jingying Zhang, Bing Liu, Tingxu Yan, Fanxing Xu, Feng Xiao, Bo Wu, Kaishun Bi, Ying Jia

**Affiliations:** 1School of Traditional Chinese Material Medical, Shenyang Pharmaceutical University, Wenhua Road 103, Shenyang 110016, China; Liaozhengzheng1989@163.com (Z.L.); JY1993@163.com (J.Z.); Liubingsy@126.com (B.L.); 2School of Functional Food and Wine, Shenyang Pharmaceutical University, Wenhua Road 103, Shenyang 110016, China; Yantingxusyphu@163.com (T.Y.); Fengxiao1996@126.com (F.X.); Bowusyphu@126.com (B.W.); 3Wuya College of Innovation, Shenyang Pharmaceutical University, Wenhua Road 103, Shenyang 110016, China; fanxing0011@163.com; 4School of Pharmacy, Shenyang Pharmaceutical University, Wenhua Road 103, Shenyang 110016, China; kaishunbisyphu@163.com

**Keywords:** a polysaccharide from *Abelmoschus esculentus* (L.) Moench, type 2 diabetes mellitus, oxidative stress, PI3K/AKT/GSK3β pathway, Nrf2 transport

## Abstract

Polysaccharide extracted from okra (*Abelmoschus esculentus* (L.) Moench), a traditional functional food, is a biologically active substance reported to possess hypoglycemic and anti-oxidative qualities. However, it is unknown which polysaccharides play a role and have the potential mechanism. This present study is to assess the possible impacts of a novel polysaccharide isolated from okra (OP) on mice fed with a high-fat diet (HFD) combined with an intraperitoneal injection (*i.p.*) of 100 mg/kg streptozotocin (STZ) twice, to induce type 2 diabetes mellitus (T2DM). We found that an eight-week administration of OP at 200 or 400 mg/kg body weight significantly alleviated the symptoms, with elevations in blood glucose, triglyceride (TG), total cholesterol (TC) and low-density lipoprotein cholesterol (LDL-C), as well as reducing high-density lipoprotein cholesterol (HDL-C), body weight, food, and water consumption. The OP treatment increased the hepatic glycogen and decreased the mussy hepatic cords and liver fibrosis in the T2DM mice. The decreases of ROS and MDA and the increases of SOD, GSH-Px and CAT in liver were observed after administration of OP. OP alleviated the T2DM characteristics through the activation of the phosphoinositide 3-kinase (PI3K)/protein kinase B (AKT)/glycogen synthase kinase 3 beta (GSK3β) pathway, and enhanced the nuclear factor erythroid-2 (Nrf2) expression and promoted Nrf2-medicated heme oxygenase-1(HO-1) and superoxide dismutase 2 (SOD2) expression. OP also relieved mitochondrial dysfunction by inhibiting NOX2 activation. Taken together, these findings suggest that a polysaccharide isolated from okra exerts anti-T2DM effects partly by modulating oxidative stress through PI3K/AKT/GSK3β pathway-medicated Nrf2 transport. We have determined that a polysaccharide possesses hypoglycemic activity, as well as its underlying mechanism.

## 1. Introduction

Diabetes is one of the serious chronic endocrine disorders in the human body worldwide, and according to statistics, more than 400 million global populations have been afflicted with it up until 2017 [1]. The number of those suffering from diabetes may presumably increase dramatically to 642 million in 2040, which will result in a huge economic burden for society, according to the prediction of the International Diabetes Federation (IDF). Diabetes is clinically classified into three types, and 90% of diabetes patients are diagnosed with type 2 diabetes mellitus (T2DM) [2]. T2DM is a well-known heterogeneous disease of protein, lipid, and carbohydrate metabolism characterized by chronic hyperglycemia or hyperlipidemia, causing inadequate insulin secretion and insulin resistance (IR) [3]. The high blood glucose level occurring in tissues with active redox cellular functions leads to the generation of reactive oxygen species (ROS), which causes oxidative stress damage particularly in the liver, pancreas, and kidney [4]. Although there are some treatment strategies for preventing and alleviating T2DM, their side effects and tolerability still cannot be ignored [5,6,7].

Recent research is focused on polysaccharides isolated from natural sources, without toxicity or side effects, which is a relatively cheaper novel method to ameliorate the development of diabetes [8]. Okra (*Abelmoschus esculentus* (L.) Moench), also known as lady’s finger or gumbo, originating from Africa and introduced to China from India in the early 20th century, is an economical vegetable crop and belongs to the mallow family [4,9]. It has a long history of treating diuretic, gastric ulcer, hepatitis, colitis, and jaundice in Chinese folk. Modern pharmacological researchers found that the okra polysaccharide could reduce blood glucose and lipid levels in different animal models. Fan and his colleagues reported that the dietary supplement of okra polysaccharides could reduce the body weight, blood glucose, and total serum cholesterol levels in high-fat diet-fed C57BL/6 mice [10]. Rhamnogalacturonan, a polysaccharide extracted from okra, has been found to have a hypoglycemic effect on streptozotocin-induced diabetic mice [11]. However, the underlying mechanism of the regulatory potential of the okra polysaccharide for anti-hyperglycemic activity is still remains to be elucidated.

The phosphoinositide 3-kinase/protein kinase B (PI3K/AKT) signaling pathway is considered to regulate various physiological processes associated with T2DM. A large number of research has found that the PI3K/AKT pathway not only promotes insulin signal transduction, but also stimulates glucose uptake in adipose tissue muscle and the liver [12,13]. On the other hand, phosphorylated AKT can also activate glycogen synthase kinase 3 beta (GSK3β; a serine/threonine protein kinase), which may promote the transfer of nuclear factor erythroid-2 (Nrf2) from the binding site of Keap1 to the nucleus, and then the downstream target genes are transactivated through relatively anti-oxidant response elements (AREs) to inhibit oxidative stress [14,15]. In summary, we hypothesize accordingly that PI3K/AKT signaling mediated Nrf2 signal plays an important role in anti-oxidative stress in T2DM. 

Based on the background, our experiment is designed to investigate anti-T2DM effects of a novel polysaccharide isolated from okra (OP) on mice with high-fat-diet feeding and streptozotocin (STZ)-induced T2DM. Moreover, the potential implications of signal kinases associated with antioxidants were also evaluated.

## 2. Results

### 2.1. Physicochemical Property Analysis of OP 

The result of the monosaccharride composition after OP derivatization is shown in Table 1. The polysaccharide content of OP is 90.93 ± 0.76%. The protein content of OP is 0.69 ± 0.03%, indicating that the protein is almost completely removed in OP. The sulfate content was 6.15 ± 0.19%. The mannose, rhamnose, glucuronic acid, galactosal acid, galactose, and arabinose of OP were 3.4:3.76:24.19:6.27:8.73:3.13, respectively. 

### 2.2. Effects of OP on Fasting Blood Glucose (FBG) Levels, Body Weight, Food, and Water Consumption

The levels of FBG that were observed in the model group were remarkably higher than the control group (*p* < 0.001). However, after the administration of OP (200 or 400 mg/kg) for eight weeks, the decreased fasting blood glucose levels were significantly reversed (*p* < 0.05, *p* < 0.001; Figure 1A). Similarly, the food and water consumption were also decreased in the treatment group compared to the model group (200 mg/kg, *p* < 0.05, *p* < 0.05 and 400 mg/kg, *p* < 0.01, *p* < 0.05) (Figure 1C,D). The body weight loss in the T2DM group was more severe than in the normal group (*p* < 0.001). This behavior could be improved after treatment with OP (200 mg/kg, *p* < 0.01 and 400 mg/kg, *p* < 0.001) (Figure 1B). 

### 2.3. Effects of OP on an Oral Glucose Tolerance Test (OGTT)

As shown in Figure 2, all of the experimental mice showed a rapid increase in blood glucose at 30 min, and later, the control group gradually decreased to normal levels within 120 min, while the blood glucose level in the model mice continued to remain the same, even until 120 min (Figure 2A), and showed the largest area under the curve (AUC) (*p* < 0.001; Figure 2B). However, supplementation with metformin and OP (400 mg/kg) resulted in decreased blood glucose within 90 min, and an AUC comparable to that of the diabetic mice (*p* < 0.05 and *p* < 0.01). 

### 2.4. Effects of OP on Organ Index 

Figure 3 show statically the increases in the index of liver (*p* < 0.001; Figure 3A), and the decrease of the index of the kidney, pancreas, and thymus that were observed in the model mice (*p* < 0.01) (Figure 3B–D), while the mice administered with OP (400 mg/kg; *p* < 0.05) were restored to the normal level. 

### 2.5. Effect of OP on the T2DM-Induced Histopathological Changes in the Liver 

The structural changes of the liver were examined by hematoxylin and eosin staining (H&E), periodic acid and Schiff staining (PAS), and masson lichun red acid reddish staining (Masson staining). In the H&E staining, we observed focal necrosis (red arrow), mussy hepatic cords (yellow arrow), and fibrosis (black arrow) in the model mice liver in the Masson staining (Figure 4A), indicating severe hepatocellular injuries in the T2DM mice. Moreover, the number of hepatic glycogen (blue arrow) in the model mice was significantly lower than that in the control mice (Figure 4B). Whereas, the OP (200 or 400 mg/kg) treatment clearly relieved those pathological changes in the liver.

### 2.6. Effects of OP on Serum Lipid Metabolism of T2DM Mice

As shown by the data in Figure 5, the triglyceride (TG), total cholesterol (TC), and low-density lipoprotein cholesterol (LDL-C) levels in the serum of the model mice were remarkably elevated compared with the blank group (*p* < 0.001). Such levels decreased after treatment with OP (Figure 5A–C). While the level of high-density lipoprotein cholesterol (HDL-C) in that of the model group significantly declined compared to the control group (*p* < 0.01), the HDL-C level of the T2DM mice supplemented with OP (400 mg/kg) increased (*p* < 0.05; Figure 5D).

### 2.7. Effects of OP on Superoxide Dismutase (SOD), Catalase (CAT), Glutathione Peroxidase (GSH-Px), and Malondialdehyde (MDA) in Serum and Liver

As shown in Table 2, the amount of SOD, CAT, and GSH-Px decreased significantly in the livers of the diabetes mice compared with the blank group (*p* < 0.01, *p* < 0.01, and *p* < 0.05, respectively). However, after eight weeks of treatment with OP (400 mg/kg) ameliorated this reduction (*p* < 0.05). The opposite results were observed regarding the level of MDA in the model mice, which was significantly increased compared with the level in the control group (*p* < 0.001), and treatment with OP (400 mg/kg) could reverse this change (*p* < 0.05).

### 2.8. Effect of OP on ROS in Liver

An increase of ROS was detected by immunofluorescence. Our data showed that the number of ROS were significantly increased in the T2DM group mice (Figure 6A,B). The OP administration could obviously reduce the number of ROSs (*p* < 0.01). 

### 2.9. Effect of OP on P-PI3K, PI3K, P-AKT, AKT, P-GSK3β, and GSK3β in Liver

As shown in Figure 7, lower level of phosphorylated P-PI3K, P-AKT, and P-GSK3β ratio were detected in model mice compared with the control mice (*p* < 0.01, *p* < 0.01, and *p* < 0.05, respectively), while supplement with OP (400 mg/kg) significantly inhibited this degradation (*p* < 0.05).

### 2.10. Effect of OP on the Expression of Nrf2, Heme Oxygenase-1 (HO-1), and Superoxide Dismutase 2 (SOD2) in the Liver

A lower expression level of nucleus Nrf2 was observed in HFD feeding and STZ induced T2DM mice compared to the control mice (*p* < 0.01), while this reduction could be reversed by metformin and OP (400 mg/kg; *p* < 0.05; Figure 8A,B). The decline in HO-1 expression could also be measured in model group and treatment with metformin and OP (400 mg/kg) could be alleviate this change (*p* < 0.05; Figure 8A,C). Accordingly, lower expression level of SOD2 was observed in model mice (*p* < 0.001), and such deregulation was evidently reversed by metformin and OP (400 mg/kg; *p* < 0.01, *p* < 0.05; Figure 8A,D). 

### 2.11. Effect of OP on the Nicotinamide Adenine Dinucleotide Phosphate Oxidases 2 (NOX2) Expression in the Liver 

Figure 9A reveals more activated NOX2 in the livers of the T2DM mice than the blank mice (*p* < 0.01; Figure 9B). Contrarily, the OP (400 mg/kg) supplemented mice showed more inactivated NOX2 (*p* < 0.05; Figure 9B). The result implied that OP can inhibit the expression of NOX2. 

## 3. Discussion

Plant polysaccharides have been explored and demonstrate significant biological activities, which account for their applications in complementary and alternative medicine. Polysaccharide, an important ingredient of *Abelmoschus esculentus* (L.) Moench, has been reported to have an excellent hypoglycemic activity and antioxidant activity [16,17], however, it is not at all certain whether OP has promoted the elimination of a high glucose impact on the body’s immune response, or whether OP has inhibited the activation of oxidative stress activation pathways. Therefore, in this paper, we mainly studied the anti-T2DM effect and the underlying mechanism of OP. HFD feeding combined with an *i.p.* lower dose of STZ showed a significant rise in the blood glucose level in both biochemical and histopathological analysis, which is conformed to the typical phenomenon of T2DM in humans [18]. This aforesaid finding is consistent with our experimental data, which measured a high blood glucose level and demonstrated pathophysiological development. Hence, this mice model would be a more suitable option to study drugs against T2DM. 

Consumptive thirst is one of the most important clinical manifestations of T2DM. Body weight loss is due to the excessive decomposition of protein and fat, which cannot be utilized by the tissues as a body energy source [19]. As a result of the blood glucose rise, plenty of water is required for the body to excrete excess glucose through the urine. The increase in food consumption is caused by a decrease in insulin sensitivity and the glucose utilizing ability of tissues, thereby leading to a long time of cell starvation. Treating T2DM is traditionally accompanied with body weight gain, as well as a decrease in food and water consumption. In this study, all of the T2DM mice clearly exhibited a gain in body weight and a decrease in food and water consumption after eight weeks of treatment. In contrast, it was observed that the body weight of the model mice were continuous decreasing, with an increase in food and water consumption (Figure 1). We speculate that this change may be related to the improvement of insulin sensitivity (Figure 2) and the promotion of hepatic glycogen synthesis, which further increases the glucose utilization by OP (Figure 7). It is known that the organ index is one of the important indicators of the histopathological changes in T2DM mice, especially in the liver, kidney, pancreas, and thymus [20]. There was a statically significant increase in the liver index, while there was a decrease in the kidney, pancreas, and thymus index of the model mice compared with the normal mice, indicating that the diabetic mice exhibited severe liver fibrosis and atrophy in the kidney, pancreas, and thymus. After treatment with OP (200 or 400 mg/kg), the liver fibrosis of the T2DM mice decreased to different degrees, meanwhile the atrophy in the kidney, pancreas, and thymus could be reversed with OP (400 mg/kg).

HFD feeding-caused T2DM in mice is generally accompanied by early symptoms of dyslipidemia, whose prominent features include elevated TG, TC, and LDL-C levels, and decreased HDL-C levels, playing a key role in the development of the pathogenesis of diabetes [21,22,23]. Consistent with the relevant studies, the result of our present study, again, shows that the long-term complement of HFD may lead to dyslipidemia. Interestingly, reduced serum TG, TC, and LDL-C levels, and raised HDL-C levels effectively occurred after the administration of OP for eight weeks, which is supportive of a previous report published by S. J. Fan et al. [10].

IR in the liver results in a disturbance of glucose homeostasis, which is another important feature of T2DM. OP and other plant polysaccharides can ameliorate insulin sensitivity by improving insulin signal defects in rodent animals [8]. In the current experiment, OP could restore insulin signals by promoting glucose utilization and effectively improving glucose tolerance in HFD-induced IR, which can also impair insulin-related intracellular signaling pathways. Apparently, the PI3K/AKT signal pathway is widely accepted as the most closely associated with insulin signal transduction [8]. The activated insulin receptor leads to the p85 subunit phosphorylation of PI3K, and then cascades the activation of AKT, which subsequently, phosphorylates GSK3β, which is the main factor in charge of the regulation of glucose metabolism in the liver. In addition, this cumulative report has indicated that HFD-induced IR can alter the PI3K/AKT/GSK3β pathway [24]. Similarly, our investigation findings also show that the dephosphorylation of PI3K, AKT, and GSK3β in the liver of HFD feeding mice obviously increased. In particular, the liver protein phosphorylation levels of PI3K, AKT, and GSK3β after the administration of OP (400 mg/kg) significantly rose.

Apart from the above effect of GSK3β, which also plays a key role in oxidative stress, accumulated evidence has indicated that oxidative stress is involved in the pathophysiology changes of T2DM [25]. The organs are exposed to long-term hyperglycemic conditions, and as a result, the generated free radicals will increase, leading to the pathogenesis of diabetes complications [26]. In fact, the defense system of antioxidant enzymes may attenuate oxidative stress by reducing ROS [27]. SOD, GSH-Px, and CAT are extensively antioxidant enzymes in vivo, and play important roles in mitigating the diseases related to oxidative stress, which may induce lipid peroxidation with MDA as a biomarker. It was observed that the activity of the antioxidant enzyme decreased while the MDA level increased in almost all of the HFD feeding and STZ-induced T2DM mice. In agreement with previous experiments, our study found that the T2DM model induced the decrease of the SOD, GSH-Px, and CAT levels in the liver, and increased the MDA level, whereas complementing with OP (400 mg/kg) could increase the levels of SOD, GSH-Px, and CAT while lowering the level of MDA in the T2DM mice (Table 2). More importantly, our study proposed a potential antioxidant mechanism by which OP protects against T2DM through the up-regulation of Nrf2 transfer to the nucleus, suggesting that methods aimed at the alleviation of the pathologic development may be effective in preventing the progression of T2DM. In addition, Nrf2 is an important transcription factor that regulates the relevant antioxidant pathway in the nucleus, which may protect redox homeostasis from oxidative damage [28]. Activated Nrf2 could scavenge the overproduced ROS via the up-regulation of HO-1 and SOD-2 expression. In our work, the T2DM model mice showed a reduced Nrf2 expression in the nucleus of the liver, indicating that the T2DM model weakened the defenses of the Nrf2-relevant antioxidant activity in vivo. Contrarily, the continuous the administration of OP for eight weeks in the T2DM mice led to the obvious up-regulation level of the nucleus Nrf2 mediated by the PI3K/Akt/GSK3β signal pathway, which is consistent with previous reports [29]. Additionally, mitochondrial dysfunction is recognized as another prominent risk of T2DM. Activated NOX2, as one of the major biomarkers of mitochondrial dysfunction, is associated with the production of ROS in STZ-induced diabetic mice [30]. In our study, it is demonstrated that the activation of NOX2 could be induced by HFD feeding and STZ, while conversely, the inactivation of NOX2 could be induced by the administration of OP (400 mg/kg), indicating that the inhibition of NOX2 activation might be a new OP therapeutic strategy. 

In conclusion, our data supports that OP demonstrates the potential for alleviating the pathologic processes of T2DM. Moreover, there is a possibility that OP may suppress the NOX2 expression and enhance the Nrf2 level in the nucleus, through the activation of the PI3K/AKT/GSK3β signal pathway in the liver, because, according to evidence, the overproduction of ROS and MDA partly decreased, while the SOD, CAT, and GSH-Px increased after supplement with OP (Figure 10). As a functional food, okra introduced into the intervention may be a new therapeutic strategy for the alleviation of oxidative stress and the reduction of T2DM risks. 

## 4. Materials and Methods 

### 4.1. Extracted and Purified Polysaccharides from Okra (OP)

The okra was obtained from the Laiyang (Shandong province, China) in August 2016. The grated dry okra was extracted with 95% ethanol under room temperature for 72 h, in order to remove pigments, small molecules, and fat-soluble substances. The air-dried residue was mixed 10 times with distilled water, and was extracted under reflux at 75 °C, three times, for 2 h each time. The filtrate was concentrated to one third of the original liquid under a vacuum at 60 °C, and then the protein was removed by the method from the previous literature [31]. The solution was precipitated by absolute alcohol to a final concentration of 80% (*v*/*v*). After being placed overnight in a refrigerator at 4 °C, the precipitates were collected by leach with a vacuum pump (R-1005, Grentwall Scientific Industrial and Trade Co., Ltd., Zhengzhou, China) to obtain the polysaccharides, and then redissolved in distilled water at 10 mg/mL, and loaded into an open DEAE-Sepharose Fast Flow column (40 × 600 mm), which was eluted in succession with dH_2_O, 0.1 M NaCl, and 0.2 M NaCl at a flow rate of 1.0 mL/min. The same fraction was combined, and then dialyzed to get the fractions OP-1 and OP-2. Based on the highest content of OP-2, it was used for further purified by an open Sepharose CL-6B gel column (4 × 60 cm), and the equivalent was eluted with 0.15 M NaCl at a flow rate of 0.1 mL/min to get OP. The total carbohydrate content in all of the experiments was determined by the previously reported methods [32]. The outflow curve of the purified OP is shown in Appendix A. The yield of OP was evaluated using the following calculation formula:The yield = OP dry weight (g)/raw material weight (g) × 100 (%)(1)

### 4.2. The Determination of Composition of OP

The total carbohydrate content was measured using the phenol–sulfuric method [33]. The protein content was determined with the Bradford method [32], and the sulfate content was measured using the barium chloride–gelatin method [33]. The monosaccharide component of OP was analyzed according to the previous method [34].

### 4.3. Animal Diets and Experimental Design

Male SPF grade C57BL/six mice (18 ± 2 g) were purchased from the Experimental Animal Center of Shenyang Pharmaceutical University, and the animal experiment was carried out under the Guideline for Animal Experimentation of Shenyang Pharmaceutical University. The protocol was approved by the Animal Ethics Committee of the institution (SCXK (Liao) 2015-001). 

Sixty-five mice were fed high-fat diets (Appendix A) for eight weeks after one week of acclimation, and then intraperitoneal injection (*i.p*.) 100 mg/kg STZ was dissolved in a pre-chilled 10% (*v*/*v*) citrate buffer (pH 4.5) at week 9 and 11. After the last injection of STZ after two weeks, 4 h-fasting blood glucose (FBG) levels in tail vein higher than 16.7 m mol/L were induced T2DM and used for further pharmacological studies. 

After induced-T2DM, the mice were divided into five groups, randomly (n = 10 per group), as follows: (1) control group + 0.5% CMC-Na; (2) model group (DM group) + 0.5% CMC-Na; (3) Met group (T2DM + 200 mg/kg Met, BW); (4) OP 200 group (T2DM + 200 mg/kg OP, BW); (5) OP 400 group (T2DM + 400 mg/kg OP, BW). All of the samples were orally administered for eight weeks. The water consumption, food intake, weight, and FBG levels were measured every two weeks. During the whole experiment, the control group was fed a normal chow diet, whereas the other groups were fed with a high-fat diet. 

### 4.4. Oral Glucose Tolerance Tests (OGTT)

After the last oral administration, the mice were fasted for 12 h (from 21:00 to 09:00 the next day), and after the oral administration of 2 g/kg of the glucose aqueous solution, the blood samples were taken from the tail vein and the blood glucose levels were measured at 30, 60, 90, and 120 min using a Jinwen glucometer (GA-3, Sinocare Biosensor Technology Co., Ltd., Changsha, China). The results were expressed as a larger area under the curve (AUC) over 120 min. The following formula was used:AUC = (BG_0min_ + BG_30min_) × 0.5h × 0.5 + (BG_30min_ + BG_120min_) × 1.5h × 0.5(2)BG_0min_, BG_30min_, and BG_120min_ represent the blood glucose levels measured at 0, 30, and 120 min, respectively.

### 4.5. Serum Collection and Liver Tissues Preparation

After OGTT, the eyeballs of the experimental mice were picked for blood and the mice were sacrificed, and then the liver was taken out quickly. Three complete liver tissues in each group were chosen randomly for tissue staining and immunohistochemistry (IHC) analyses, and they were stored in 4% paraformaldehyde, and one complete liver tissue in each group was immunofluorescence detection rapidly. The remaining livers were used for Western blot (WB) analysis and enzyme-linked immunosorbent assay (ELLSA) kit and Western blot (WB) analysis, and were stored at −80 °C until the biochemical analysis.

### 4.6. Biochemical Analysis

#### 4.6.1. Enzyme linked immunosorbent assay (ELISA) Assay Kits

The concentrations of the serum TC, TG, LDL-C, and HDL-C, as well as the SOD, GSH-Px, CAT, and MDA in the liver were measured by a specific ELISA kit (Jiancheng, Jiangsu, China), according to the instruction manual from the manufacturer.

#### 4.6.2. Western Blot

The determination of the total protein of the liver as described previously [20]. Briefly, the livers were homogenized by an ultrasonic crusher in an ice-water bath phenylmethanesulfonyl fluoride (PMSF) and radio immunoprecipitation assay (RIPA) buffer (1:100), and were centrifuged (H1850R, Xiangyi Centrifuge Instrument Co., Ltd., Changsha, China) at 15,000× *g* for 15 min at 4 °C, and then the clarifying liquid was collected. The protein contents were determined using the bicinchoninic acid (BCA) protein assay kit (MA 0082; Dalian meilune Bioengineering research institute, China). Then, the protein was fractionated by sodium dodecyl sulfate polyacrylamide gel electrophoresis (SDS-PAGE) gel using 10% or 12% resolving gel. Next, the protein samples were transferred by electrophoresis onto the polyvinylidene fluoride (PVDF) membrane. Later, the membrane was blocked by a 7.5% skim milk solution at room temperature for 2 h. The primary antibodies of PI3Kp85 (1:1000, Abcam, Cambridge, UK), p-PI3Kp85 (1:1000, Abcam, Cambridge, UK), AKT (1:1000, CST, Boston, USA), p-AKT (1:1000, Abcam, Cambridge, UK), GSK3β (1:1000, CST, Boston, USA), p-GSK3β (1:1000, CST, Boston, USA), Nrf2 (1:500, Proteintech, Chicago, USA), HO-1(1:500, Proteintech, Chicago, USA), SOD2 (1:1000, CST, Boston, USA), and anti-β-actin antibody (1:3000, CST, Boston, USA) were dissolved in 7.5% fat-free milk as a binding target protein, at 4 °C for 12 h. After three washes of the membrane in TBST, for 5 min each time, the HRP-conjugated anti-rabbit antibody was incubated at 25 °C for 60 min, and then later was washed again three times with TBST. The bands of proteins were immunolabeled by an ELC detection kit in a dark room (Dalian meilune Bioengineering research institute, China). The protein bands were obtained through film. 

#### 4.6.3. Immunohistochemical (IHC) and Immunofluorescence Analyses

The livers were embedded in paraffin and then cut into 3 μm transverse slices. The slices were placed in a roaster (RM2016, Leica Instrument Co., Ltd., Shanghai, China) at 60 °C for 2 h. Next, the livers were dewaxed three times in xykene, for 15 minutes each time, and then the residual liquid was removed. The slice was washed with absolute ethanol and deionized water, and then incubated for 10 min at 25 °C in 100 μL of 3% H_2_O_2_ so as to block the activation of the endogenous peroxidase. Then, the slices were washed in phosphate buffer saline (PBS) for 10 min, followed by incubation in a 5% serum blocking solution at 25 °C for 5 min. Next, the slices were incubated with the primary antibody for NOX2 (1:200, Abcam, Cambridge, UK) diluent at 4 °C for 12 h. After that, the slices were washed with PBS for 10 min and were then incubated with the secondary antibodies at 37 °C for 30 min. Subsequently, the slices were washed with PBS again for 10 min, and incubated with a strept avidin-biotin complex (SABC) reagent at a constant temperature of 37 °C for 30 min, and then washed with PBS for 10 min. After the residual PBS was removed, 50 μL of DAB was loaded onto the slices they were dehydration successively with 75% ethanol, 85% ethanol, 85% ethanol, 95% ethanol, and 100% ethanol, for 5 min each. The slices were placed in xylene for 5 min, three times, and then the residual liquid was removed. The slices were covered in glasses with neutral gum as an adhesive. The photomicrographs were obtained by a computer with an Axiocam 50 camera, and then analyzed by Image-Pro Plus software 6.0 (Media Cybernetics Co., Ltd., Rockville, MD, USA). 

Some sections were incubated with 10% goat serum containing 0.2% Triton × 100 at 4 °C overnight to block the nonspecific expression. The slices were incubated with a primary antibody for ROS in 2% serum overnight at 4 °C, and the next day, the slices were washed with PBS for 5 min, three times, and then incubated in rabbit anti-goat IgG diluent for 1 h at 25 °C. The slices were washed again with PBS for 5 min, three times, and were covered with DAPI or Flurorescence microscope by a computer with a video camera (Axioscope A1, Dresden, Germany). 

#### 4.6.4. Histopathological Examination

The sections were obtained from the steps from Section 2.6, and then the sections stained with different staining solutions (H&E staining—hematoxylin and eosin; Masson staining—Masson lichun red acid reddish; PAS staining—periodic acid and Schiff).

### 4.7. Statistical Analysis

All of the data are presented as mean ± standard deviation (SD), and were evaluated by one-way analysis of variance (ANOVA) and the Tukey’s test. The difference was considered to be statistically significant if *p* < 0.05. All of the statistical analyses were carried out by using SPSS software 19.0 (International Business Machines Co., Armonk, NY, USA). 

## Figures and Tables

**Figure 1 molecules-24-01906-f001:**
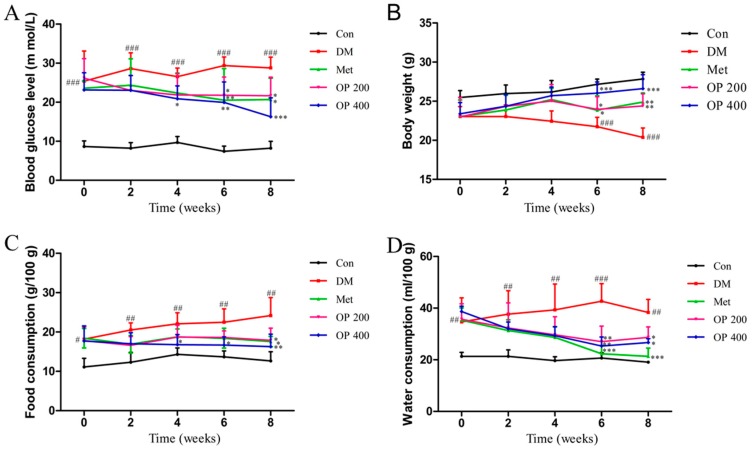
Effects of okra (OP) on fasting blood glucose level (**A**), body weight (**B**), food consumption (**C**) and water consumption (**D**). Data are expressed as means ± standard deviation (SD; n = 10). Con—control group treated with 0.5% sodium carboxyl methyl cellulose (CMC-Na) per day; DM—type 2 diabetes melltius (T2DM) treated with 0.5% CMC-Na per day; Met—T2DM mice treated with 200 mg/kg metformin per day; OP 200—T2DM mice treated with 200 mg/kg OP per day; OP 400—T2DM mice treated with 400 mg/kg OP per day. Data are expressed as means ± SD (n = 10), ^#^
*p* < 0.05, ^##^
*p* < 0.01, ^###^
*p* < 0.001 vs. control, * *p* < 0.05, ** *p* < 0.01, *** *p* < 0.001 vs. model.

**Figure 2 molecules-24-01906-f002:**
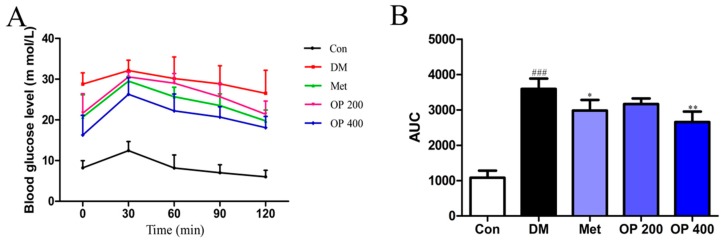
Effects of OP on glucose tolerance test (**A**) and area under curve (AUC) (**B**) in the oral glucose tolerance test (OGTT) mice. Each value is mean ± SD of eight mice in experimental groups. ^###^
*p* < 0.001 vs. control, * *p* < 0.05, ** *p* < 0.01 vs. model.

**Figure 3 molecules-24-01906-f003:**
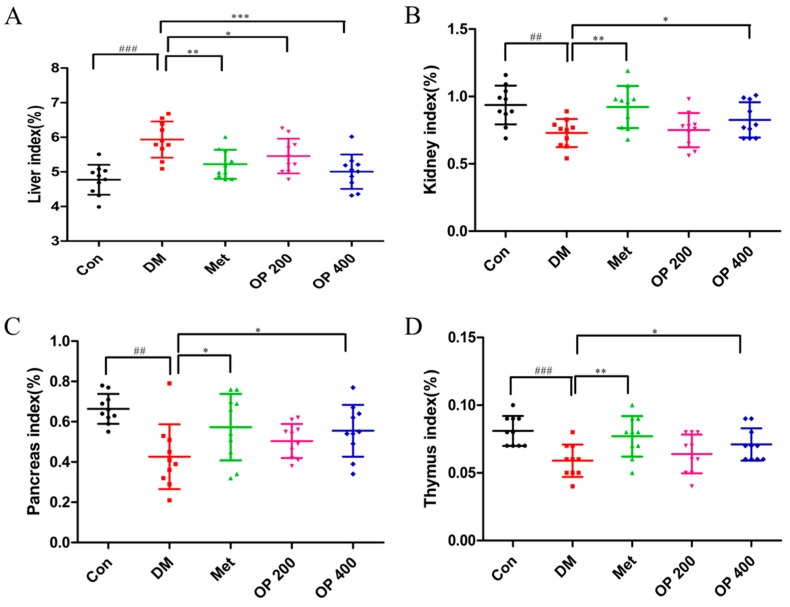
Effects of OP on organ index of whole experimental mice. (**A**) Liver index; (**B**) kidney index; (**C**) pancreas index; (**D**) thymus index. Data represent means ± standard error of the mean (SEM; n = 10). ^##^
*p* < 0.01, ^###^
*p* < 0.001 vs. control, * *p* < 0.05, ** *p* < 0.01, *** *p* < 0.001 vs. model.

**Figure 4 molecules-24-01906-f004:**
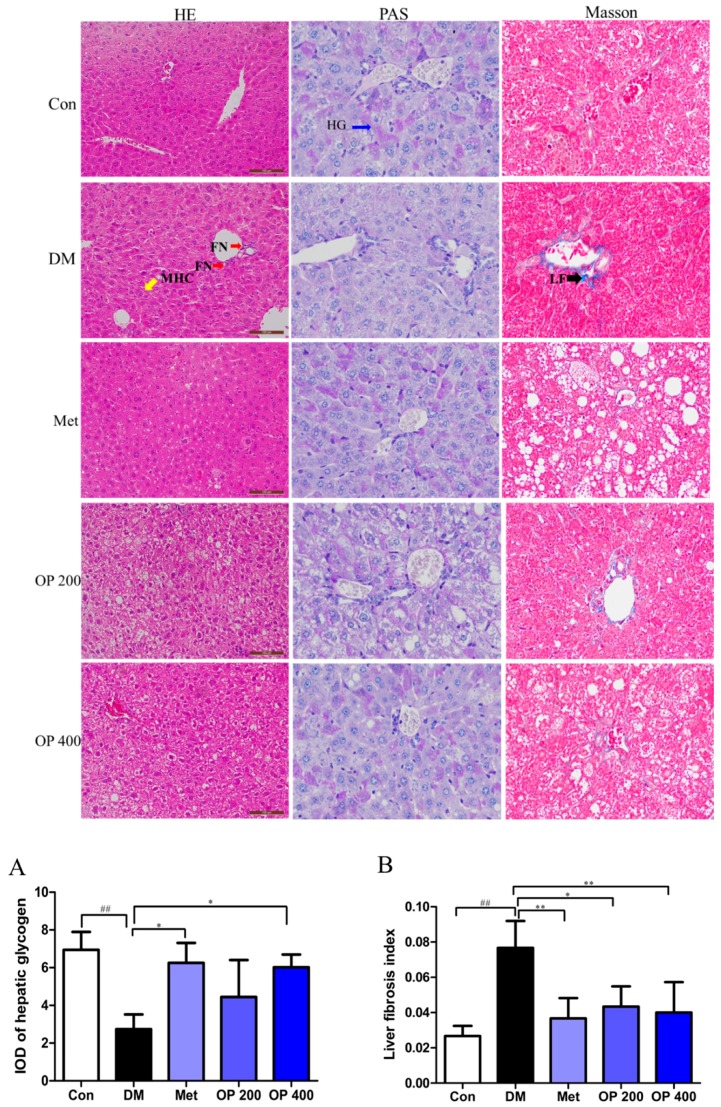
Effects of OP on the T2DM-induced histopathological changes in liver (hematoxylin and eosin (H&E) staining, 200×; periodic acid and Schiff staining (PAS) and Masson staining, 400×). FN—focal necrosis; MHC—mussy hepatic cords; HG—hepatic glycogen; LF—liver fibrosis. (**A**) Integral optical density (IOD) of the hepatic glycogen. (**B**) IOD of the liver fibrosis. The analysis of IOD by Image-Pro Plus software 6.0. Results are represented as mean ± SD (n = 3). ^##^
*p* < 0.01 vs. control, * *p* < 0.05, ** *p* < 0.01 vs. model.

**Figure 5 molecules-24-01906-f005:**
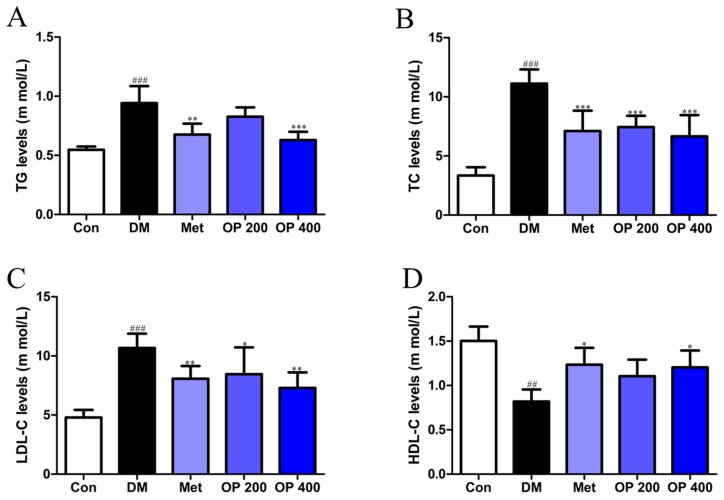
Effects of OP on (**A**) serum triglyceride (TG), (**B**) total cholesterol (TC), (**C**) low density lipoprotein cholesterol (LDL-C), and (**D**) high-density lipoprotein cholesterol (HDL-C) of experimental mice. Data represent means ± SEM (n = 10). ^##^
*p* < 0.01, ^###^
*p* < 0.001 vs. control, * *p* < 0.05, ** *p* < 0.01,*** *p* < 0.001 vs. model.

**Figure 6 molecules-24-01906-f006:**
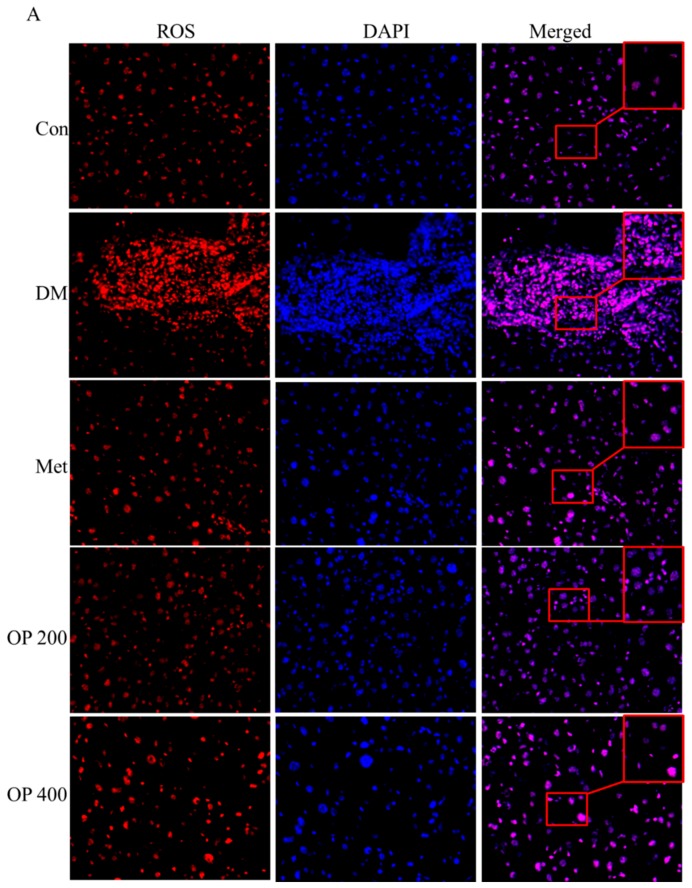
Effect of OP on reactive oxygen species (ROS) in the liver. (**A**) Liver were stained with specific antibodies against ROS (red). Nuclei were stained with 4’,6-diamidino-2-phenylindole (DAPI) (blue). (**B**) The IOD of ROS. Results are expressed as mean ± SEM (n = 3). ^###^
*p* < 0.001 vs. control, * *p* < 0.05, ** *p* < 0.01 vs. model.

**Figure 7 molecules-24-01906-f007:**
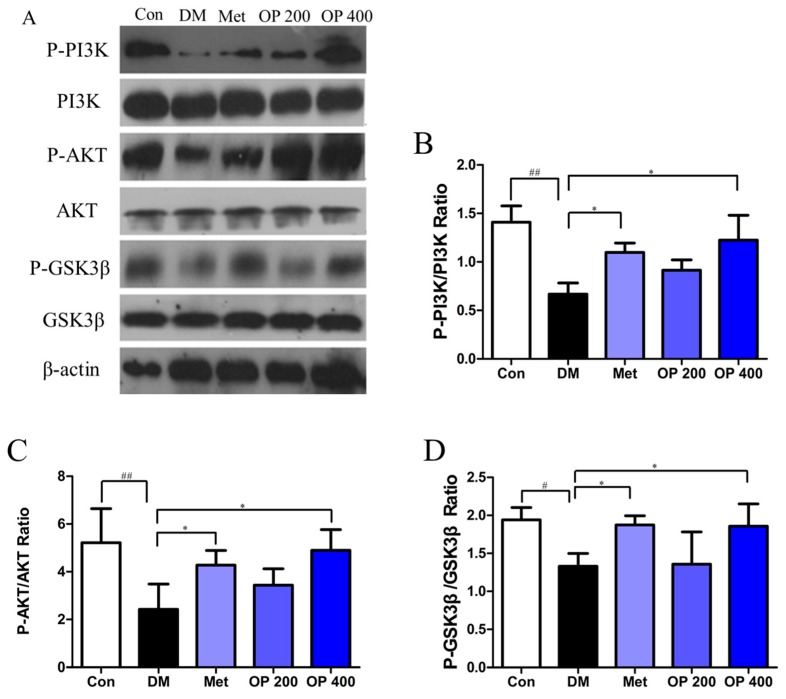
Effect of OP on Phosphorylation-phosphoinositide 3-kinase (P-PI3K), PI3K, P-protein kinase B (AKT), AKT, P-glycogen synthase kinase 3 beta (GSK3β), and GSK3β in liver. (**A**) The protein expression analysis from liver of experimental mice using P-PI3K, PI3K, P-AKT, AKT, P-GSK3β, and GSK3β antibodies by Western blot. The data of densitometric analysis of (**B**) P-PI3K/PI3K, (**C**) P-AKT/AKT and (**D**) P-GSK3β/ GSK3β. Results are expressed as mean ± SEM (n = 3). ^#^
*p* < 0.05, ^##^
*p* < 0.01 vs. control, * *p* < 0.05 vs. model.

**Figure 8 molecules-24-01906-f008:**
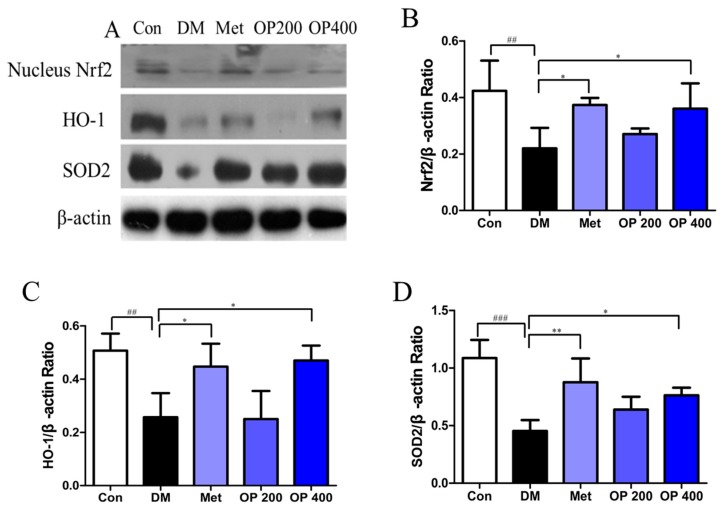
(**A**) Effect of OP on the expression of nuclear factor erythroid-2 (Nrf2), heme oxygenase-1 (HO-1), and superoxide dismutase 2 (SOD2) in the liver. The protein expression analysis from liver of experimental mice using Nrf2, HO-1, and SOD2 antibodies. The data of the densitometric analysis of (**B**) Nrf2/β-actin, (**C**) HO-1/β-actin, and (**D**) SOD2/β-actin. Results are expressed as mean ± SEM (n = 3). ^##^
*p* < 0.01, ^###^
*p* < 0.001 vs. control, * *p* < 0.05, ** *p* < 0.01 vs. model.

**Figure 9 molecules-24-01906-f009:**
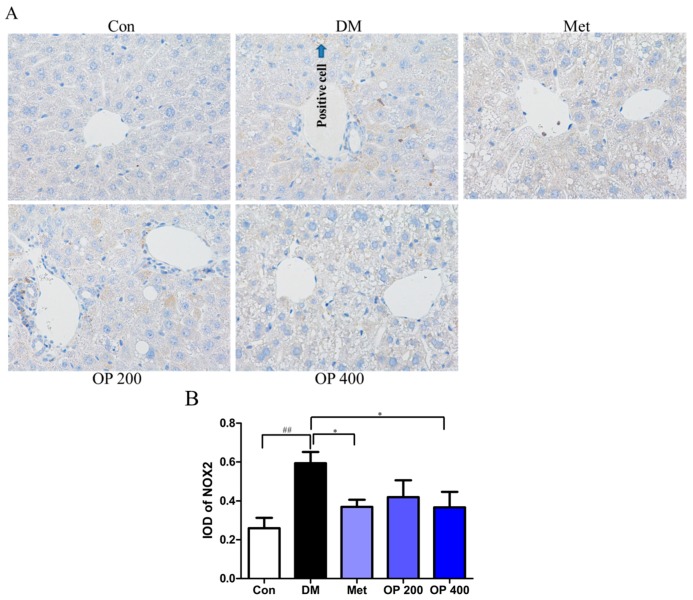
Effect of OP treatment on the T2DM-induced nicotinamide adenine dinucleotide phosphate oxidases 2 (NOX2) expression in the liver (Diaminobenzidine (DAB) staining, 200×). (**A**) Representative photographs were shown for the mouse livers in the different groups. (**B**) IOD of the NOX2 by Image-Pro Plus software 6.0 analysis. Results are represented as mean ± SD (n = 3). ^##^
*p* < 0.01 vs. control, * *p* < 0.05 vs. model.

**Figure 10 molecules-24-01906-f010:**
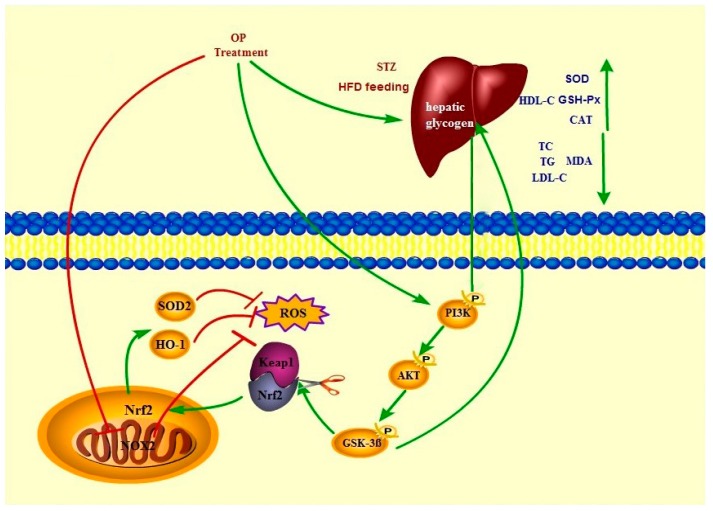
Proposed OP anti T2DM-like pathways.

**Table 1 molecules-24-01906-t001:** Physicochemical characterization of okra (OP).

Yield (%) ^a^	Polysaccharide (%)	Protein (%)	Sulfate (%)	Mannose	Rhamnose	Glucuronic Acid	Galactosal Acid	Galactose	Arabia
0.59	90.93 ± 0.76	0.69 ± 0.03	6.15 ± 0.19	3.4 ± 0.02	3.76 ± 0.01	24.19 ± 0.02	6.27 ± 0.01	8.73 ± 0.02	3.13 ± 0.02

The % is related to dry weight. Data are expressed as means ± standard deviation (SD; n = 3).

**Table 2 molecules-24-01906-t002:** Effects of OP on the activity of antioxidant enzymes in serum and liver.

Parameter	Con	DM	Met	OP	
				200 mg/kg	400 mg/kg
SOD (U/mg prot)	306.89 ± 73.9	201.87 ± 47.1 ^##^	284.54 ± 19.5 *	252.51 ± 46.5	274.18 ± 24.1 *
CAT (U/mg prot)	65.13 ± 11.1	47.63 ± 7.79 ^##^	57.80 ± 6.60 *	50.32 ± 3.09	57.09 ± 6.91 *
GSH-Px (U/mg prot)	562.53 ± 151	411.82 ± 26.4 ^#^	526.63 ± 59.1 *	461.05 ± 90.2	530.08 ± 45.1 *
MDA (n mol/mg prot)	2.71 ± 0.44	4.27 ± 0.31 ^###^	3.17 ± 0.55 **	4.02 ± 0.71	3.48 ± 0.45 *

Data are expressed as means ± standard deviation (SD) (n = 6). ^#^
*p* < 0.05, ^##^
*p* < 0.01, ^###^
*p* < 0.001 vs. control. * *p* < 0.05 vs. model. SOD—superoxide dismutase; CAT—catalase; GSH-Px—glutathione peroxidase; MDA—malondialdehyde.

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
