# Peer review of "Polysaccharide from Okra (Abelmoschus esculentus (L.) Moench) Improves Antioxidant Capacity via PI3K/AKT Pathways and Nrf2 Translocation in a Type 2 Diabetes Model"

_molecules, 2019, doi:10.3390/molecules24101906_

Round 1
Reviewer 1 Report
This is an interesting manuscript that describe possible impacts of a novel polysaccharide isolated from okra (Abelmoschus esculentus (L.) Moench) (OP) on mice with high fat diet (HFD) feeding combined with 19 intraperitoneal injection (i.p.) of 100 mg/kg streptozotocin (STZ) twice induce type 2 diabetes 20 mellitus (T2DM).
Major comment;
Glucuronic acid seem to be the major content of OP (Table 1). Which content of OP contribute the anti-T2DM effects partly by modulating oxidative stress through PI3K/AKT/GSK3β pathway-medicated Nrf2 transport?
Why the supplementation with metformin (Figure 2) did not control blood sugar level as control in the OGTT mice?
How the define the index of liver, index of kidney etc?
Author might explain the reason that HDL-C level of T2DM mice of those supplementation with OP (400 mg/kg) instead of OP (200 mg/kg) (Fig 5D)
As author mention that "Insulin resistance (IR) in the liver results in disturbance of glucose homeostasis". Why did not use insulin to control blood sugar level then perform experiment to OP contribute the anti-T2DM effects partly by modulating oxidative stress through PI3K/AKT/GSK3β pathway-medicated Nrf2 transpor?
Overall, the manuscript is a good writing focus on the a novel polysaccharide isolated from OP that exerts anti-T2DM effects partly by modulating oxidative stress through
PI3K/AKT/GSK3β pathway-medicated Nrf2 transport.
Author Response
Point 1: Glucuronic acid seems to be the major content of OP (Table 1). Which content of OP contribute the anti-T2DM effects partly by modulating oxidative stress through PI3K/AKT/GSK3β pathway-medicated Nrf2 transport?
Response 1: Thank you for the referee’s suggestion. It’s my mistake for the manuscript is not clearly stated. I have already added an introductive sentence in the manuscript as follow: “The result of the monosaccharride composition after OP derivatization was shown in Table 1.” OP is a polysaccharide. Glucuronic acid is the content obtained after derivatization of OP, and the identification material of OP is in the supplementary material. Furthermore, some studies have reported that a polysaccharide regulate oxidative stress through the PI3K/AKT/GSK3β signaling pathway-medicated Nrf2 transport [1-3]. So, in this experiment, OP contributes the anti-T2DM effects partly by modulating oxidative stress through PI3K/AKT/GSK3β pathway-medicated Nrf2 transport. Thank you.
Point 2: why the supplementation with metformin (Figure 2) did not control blood sugar level as control in the OGTT mice?
Response 2: Thank you for the referee’s suggestion. There are some reports metformin did not control blood sugar level as control in the OGTT mice [4,5]. Consistent with these studies, the result of our present study again shown that metformin did not control blood sugar level as control in the OGTT mice. The different results may be due to the fact that STZ does not have the same damage to the β cells, or the formula of the feed is different, etc. Thank you.
Point 3: How the define the index of liver, index of kidney etc?
Response 3: Thank you for the referee’s kind suggestion. The calculation formula for index of liver, kidney, pancreas and thymus as follows:
Organ index (%) = Wet weight of organs/Body weight × 100%
Point 4: Author might explain the reason that HDL-C level of T2DM mice of those supplementation with OP (400 mg/kg) instead of OP (200 mg/kg) (Fig 5D).
Response 4: Thank you for the referee’s kind suggestion. HDL-C is commonly known as "vascular scavenger". It can convert cholesterol into bile acids or directly from the intestines through bile. Within a certain range, HDL-C can effectively protect blood vessels. So, supplementation with OP (400 mg/kg) raises HDL-C level to better protect blood vessels. Administration of OP (200 mg/kg) can also increase the HDL-C levels, but no significant difference. The difference in result may be due to the higher concentration of OP, the effect is more significant. Thank you.
Point 5: As author mention that "Insulin resistance (IR) in the liver results in disturbance of glucose homeostasis". Why did not use insulin to control blood sugar level then perform experiment to OP contribute the anti-T2DM effects partly by modulating oxidative stress through PI3K/AKT/GSK3β pathway-medicated Nrf2 transpor?
Response 5: Thank you for the referee’s kind suggestion. Insulin is a peptide drug. Once insulin is taken orally, it will be broken down into amino acids by the digestive juice in the intestines, losing the integrity of insulin and affecting its hypoglycemic effect. Moreover, there are no oral insulin dosage forms available on the market. In addition, the experiment was designed to study the hypoglycemic activity of polysaccharides after oral administration. In order to the parallel experiments, we chose oral metformin as a positive drug in this experiment. Some literature was also used metformin as a positive drug to develop other drugs that improve insulin resistance in vitro and in vivo experiments [3, 6, 7]. Base on the above reasons, we choose the metformin orally to control blood sugar level then perform experiment. Thank you.
References
1. Li, W.; Cai, Z.N.; Mehmood, S.; Wang,Y .; Pan, W.J.; Zhang, W.N.; Lu, Y.M.; Chen,Y. Polysaccharide FMP-1 from Morchella esculenta attenuates cellular oxidative damage in human alveolar epithelial A549 cells through PI3K/AKT/Nrf2/HO-1 pathway. Int. J. Biol. Macromol. 2019, 120, 865-875.
2. Gao, Q.H.; Fu, X.Y.; Zhang, R.; Wang, Z.S.; Guo, M.Z. Neuroprotective effects of plant polysaccharides: A review of the mechanisms. Int. J. Biol. Macromol. 2018 106,749-754.
3. Yang, Y.; Li, W.; Li, Y.; Wang, Q.; Gao, L.; Zhao, J.J. Dietary Lycium barbarum Polysaccharide Induces Nrf2/ARE Pathway and Ameliorates Insulin Resistance Induced by High-Fat via Activation of PI3K/AKT Signaling. Oxid. Med. Cell. Longev. 2016, DOI: 10.1155/2014/145641.
4. Jiao, Y.K.; Wang, X.Q.; Jiang, X.; Kong, F.S.; Wang, S.M.; Yan, C.Y. Antidiabetic effects of Morus alba fruit polysaccharides on high-fat diet and streptozotocin-induced type 2 diabetes in rats. J. Ethnopharmacol. 2017, 199, 119-127.
5. Hu, F.D.; Li, X.D.; Zhao, L.G.; Feng, S.L.; Wang, C.M. Antidiabetic properties of purified polysaccharide from Hedysarum polybotrys. Can. J. Physiol. Pharmacol. 2010, 88, 64-72.
6. Lei, Y.Y.; Gong, L.L.; Tan F.G.; Liu, Y.X.; Li, S.S.; Shen, H.W.; Zhu, M.; Cai, W.W.; Xu, F.; Hou, B.; Zhou, Y.T.; Han, H.X.; Qiu, L.Y.; Sun, H.J. Vaccarin ameliorates insulin resistance and steatosis by activating the AMPK signaling pathway. Eur. J. Pharmacol. 2019, 851, 13-24.
7. Madiraju, A.K.; Qiu, Y.; Perry, R.J.; Rahimi, Y.; Zhang, X.M.; Zhang, D.Y.; Camporez, J.P.G.; Cline, G.W.; Butrico, G.M.; Kemp, B.E.; Casals, G.; Steinberg, G.R.; Vatner, D.F.; Petersen, K.F.; Shulman, G.I. Metformin inhibits gluconeogenesis via a redox-dependent mechanism in vivo. Nat. Med. 2018, 24(9), 1384-1394.

Reviewer 2 Report
The paper is well structured, and the conclusions were confirmed by the experiment results. The paper seems worthy of publication in “Molecules” however the authors are invited to consider the following comments:
*/ The abstract: usual sections defined in a structured abstract are the Background, Methods, Results, and Conclusions. Please add an introductive sentence and conclusion to improve the abstract.
*/ Table 1: please add table footnote: the % is related to dry weight!? are The data presented in triplicate? Please express sugar as means ± SD.
*/ please explain why the data were expressed as means ± SD (1, 4 and 9) and means ± S.E.M (fig. 3, 5, 6, 7 and 8). Or make it uniform in the whole manuscript
*/ in the manuscript all abbreviations should be explained at first citation, for example, H.E Hematoxylin eosin. please revise this in the whole manuscript
*/ in figure 10: please specify in the pathway which ones are the achievement of this study and which are the data from the literature.
*/ please revise the manuscript for several syntax and grammar errors.
The attached file may help to correct some mistakes however I advise to revise the paper language by an expert.

Author Response
Point 1: The abstract: usual sections defined in a structured abstract are the Background, Methods, Results, and Conclusions. Please add an introductive sentence and conclusion to improve the abstract.
Response 1: Thank you for the referee’s kind suggestion. I have added an introductive sentence as follow: “Polysaccharide extracted from okra (Abelmoschus esculentus (L.) Moench), a traditional function food, is a biologically active substance reported to possess hypoglycemic and anti-oxidative. However, it is unknown which polysaccharide play the role and potential mechanism” and the conclusion as follow: “We have determined that a polysaccharide possesses hypoglycemic activity and its underlying mechanism” in the abstract. Thanks again for your advices.
Point 2: Table 1, please add table footnote: the % is related to dry weight!? Are The data presented in triplicate? Please express sugar as means ± SD
Response 2: Thank you for the referee’s kind suggestion. The Table 1 footnote was added. The data presented in triplicate. The sugar was expressed as means ± SD. Thanks again for your advices.
Point 3: Please explain why the data were expressed as means ± SD (1, 4 and 9) and means ± S.E.M (Fig. 3, 5, 6, 7 and 8). Or make it uniform in the whole manuscript.
Response 3: Thank you for the referee’s kind suggestion. I apologize to you for my mistakes. The data expression has been made uniform. Thanks again for your advices.
Point 4: All abbreviations should be explained at first citation, for example, H.E. (Hematoxylin eosin). Please revise this in the whole manuscript.
Response 4: Thank you for the referee’s kind suggestion. It’s my mistakes. I have explained the abbreviations at first citation in the whole manuscript. Thank you.
Point 5: in figure 10: please specify in the pathway which ones is the achievement of this study and which are the data from the literature.
Response 5: Figure 10 was a summary of the present experimental data except that the levels of Nrf2 and Kepa1 in cytoplasm were not detected. Furthermore, there are many literatures on the levels of Keap1 and Nrf2 in the cytoplasm [1-3].
Point 6: Please revise the manuscript for several syntax and grammar errors.
The attached file may help to correct some mistakes however I advise to revise the paper language by an expert.
Response 6: Thank you for the referee’s kind suggestion. The manuscript’s syntax and grammar errors have revised by a native English expert as follows:
1. Page 1, line 37, “Diabetes is one of serious chronic endocrine disorder in the world, and it has now afflicted more than 400 million global populations in 2017” changed to “Diabetes is one of the serious chronic endocrine disorders in the human body worldwide, and according to the statistics more than 400 million global populations had been afflicted with it up to 2017.”
2. Page 1, line 39, “It prevalence will…” changed to “The number of those suffering from diabetes may presumably…”
3. Page 1, line 40, “and will bring…” changed to “which will result in”
4. Page 1, line 40, “according to the International Diabetes Federation (IDF) predicts” changed to “according to the prediction of International Diabetes Federation (IDF).”
5. Page 1, line 42, “have…” changed to “are diagnosed with…”
6. Page 1, line 45, “occurs…” changed to “occurring”
7. Page 1, line 45, “leading…” changed to “leads”
8. Page 1, line 46, “reactive oxygen species (ROS) to cause oxidative stress” changed to “reactive oxygen species (ROS), which causes oxidative stress”
9. Page 1, line 47, “prevent and alleviate” changed to “preventing and alleviating.”
10. Page 1, line 47, “the side effects” changed to “their side effects”.
11. Page 2, line 49, “Polysaccharide isolated from natural sources with no toxicity and side effect that is relatively cheaper to use as a novel method to ameliorate the development of diabetes are the central issue of recent research” changed to “Recent researches are focused on polysaccharide isolated from natural sources without toxicity and side effect, which is relatively cheaper as a novel method to ameliorate the development of diabetes.”
12. Page 2, line 52, “is an economical vegetable crop belonging to the mallow family and originating from Africa and it introduced to China from India in the early 20th century” changed to “originating from Africa and introduced to China from India in the early 20th century, is an economical vegetable crop and belong to the mallow family ”
13. Page 2, line 53, “It has a long history for…” changed to “It has a long history of…”
14. Page 2, line 54, “The modern…” changed to “Modern…”
15. Page 2, line 56, “Fan and his colleagues reported that dietary supplement of okra polysaccharides could lower the body weight…” changed to “Fan and his colleagues reported that the dietary supplement of okra polysaccharides could reduce the body weight…”
16. Page 2, line 58, “a polysaccharide extract from okra have hypoglycemic effect in…” changed to “a polysaccharide extracted from okra has been found hypoglycemic effect on…”
17. Page 2, line 60, “anti-hyperglycemic activity is not elucidated” changed to “anti-hyperglycemic activity is stills remain to be elucidated”
18. Page 2, line 63, “A large number of research finding that…” changed to “A large number of research found that…”
19. Page 2, line 65, “in the adipose tissue muscle and liver” changed to “in adipose tissue muscle and liver”
20. Page 2, line 66, “GSK3β (a serine/threonine protein kinase) to promote Nrf2 (nuclear factor erythroid-2) from binding site of Keap1 transferred to nucleus, and then transactivates its downstream target genes through…” changed to “GSK3β (a serine/threonine protein kinase), which may promote the transfer of Nrf2 (nuclear factor erythroid-2) from binding site of Keap1 to nucleus, and then the downstream target genes are transactivated ”
21. Page 2, line 69, “We hypothesis PI3K/AKT signaling mediated Nrf2 signal is play important role in anti-oxidative stress in T2DM” changed to “we hypothesize accordingly that PI3K/AKT signaling mediated Nrf2 signal plays an important role in anti-oxidative stress in T2DM”
22. Page 2, line 71, “we design this experiment…” changed to “our experiment is designed…”
23. Page 2, line 72, “in high-fat-diet feeding and streptozotocin (STZ)-induced T2DM mice” changed to “on mice with high-fat-diet feeding and streptozotocin (STZ)-induced T2DM”
24. Page 2, line 73, “and the potential implication…” changed to “Besides, the potential implications…”
25. Page 2, line 78, “The polysaccharide content of OP were…” changed to “The polysaccharide content of OP was…”
26. Page 2, line 88, “Food and water consumption…” changed to “Food and water consumption…”
27. Page 3, line 101, “gradully” changed to “gradually”
28. Page 4, line 119, “liver were examined…” changed to “liver was examined…”
29. Page 6, line 144, “Liver” changed to “liver”
30. Page 6, line 146, “diatebes” changed to “diabetes”
31. Page 6, line 151, “Liver” changed to “liver”
32. Page 7, line 156, “adminstration” changed to “administration”
33. Page 10, line 201, “Plant polysaccharides have been explored to show important biological activities, which account for its…” changed to “Plant polysaccharides have been explored and demonstrate significant biological activities, which account for the applications in…”
34. Page 10, line 202, “Polysaccharides are the important ingredient…” changed to “Polysaccharide, an important ingredient…”
35. Page 10, line 203, “have been reported to have good hypoglycemic activity…”changed to “has been reported to have excellent hypoglycemic activity…”
36. Page 10, line 204, “but we were not sure whether OP promote the elimination of high glucose to body’s immune response or whether OP inhibit the activation of oxidative stress activation pathways to play the role” changed to “however, it is not at all certain whether OP has promoted the elimination of high glucose impact on body’s immune response or whether OP has inhibited the activation of oxidative stress activation pathways”
37. Page 10, line 206, “So, in present paper” changed to “Therefore, in this paper”
38. Page 10, line 207, “under” changed to “underlying”
39. Page 10, line 209, “which has been conformed to…” changed to “which is conformed to…”
40. Page 10, line 210, “This is consistent with our experimental data that high blood glucose level and pathophysiological development was measured” changed to “This aforesaid finding is consistent with our experimental data that measured the high blood glucose level and demonstrated pathophysiological development”
41. Page 10, line 211, “Therefore, this mice model would be a more suitable for studying drugs against T2DM” changed to “Hence, this mice model would be a more suitable option to study drugs against T2DM”
42. Page 10, line 213, “Consumptive thirst is the most important…” changed to “Consumptive thirst is one of the most important…”
43. Page 10, line 214, “Body weight loss was due to the excessive decomposition of protein and fat in a condition of unavailability that for utilization by the tissue as a body energy source” changed to “Body weight loss is due to the excessive decomposition of protein and fat which cannot be utilized by the tissues as a body energy source”
44. Page 10, line 215, “After the blood glucose rises, the body needs to drink plenty of water to…” changed to “As a result of blood glucose rise, plenty of water is requires for the body to…”
45. Page 10, line 217, “and the ability of tissues to utilize glucose…” changed to “and the glucose utilizing ability of tissues… ”
46. Page 10, line 217, “thereby causing cells to be starved for a long time” changed to “thereby leading to a long time of cell starvation”
47. Page 10, line 218, “Treating T2DM is traditionally accompanied with the body weight gain, food and water consumption decrease” changed to “Treating T2DM is traditionally accompanied with body weight gain, as well as decrease in food and water consumption”
48. Page 11, line 219, “there were all the T2DM mice…” changed to “all the T2DM mice…”
49. Page 11, line 220, “However, body weight of model mice was continuous decreasing. Conversely, the increased food and water consumption were obviously observed in model mice” changed to “In contrast, it was observed that the body weight of model mice was continuous decreasing, with an increase in food and water consumption”
50. Page 11, line 223, “and promotion of hepatic glycogen synthesis to increase glucose utilization by the OP” changed to “and promotion of hepatic glycogen synthesis, which further increases glucose utilization by the OP”
51. Page 11, line 224, “The organ index is one of…” changed to “As is known that organ index is one of…”
52. Page 11, line 226, “increase in liver index, as well as decrease in…”changed to “increase in liver index, while a decrease in…”
53. Page 11, line 227, “with normal mice, indicating the diabetic mice exhibited severe liver fibrosis and renal, pancreas and thymus atrophy” changed to “with normal mice, indicating that the diabetic mice exhibited severe liver fibrosis and atrophy in renal, pancreas and thymus”
54. Page 11, line 229, “as well as the OP (400 mg/kg) could be reverse the atrophy in renal, pancreas and thymus” changed to “meanwhile the atrophy in renal, pancreas and thymus could be reversed with OP (400 mg/kg) ”
55. Page 11, line 232, “dyslipidemia. Dyslipidemia plays a key role for the development of pathogenesis of diabetes. Elevated TG, TC, LDL-C levels and decreased HDL-C level are prominent feature of dyslipidemia.” changed to “dyslipidemia, whose prominent features include elevated TG, TC, LDL-C levels and decreased HDL-C level, playing a key role in the development of pathogenesis of diabetes”
56. Page 11, line 234, “Consistent with other studies, the result of our present study again shown that the long-term complement of HFD led to dyslipidemia” changed to “Consistent with relevant studies, the result of our present study again shows that long-term complement of HFD may lead ”
57. Page 11, line 235, “administration of OP for 8 weeks was effective to reduce serum TG, TC, LDL-C levels and raised HDL-C level, and our results support for a previous report by S. J. Fan et al” changed to “reduced serum TG, TC, LDL-C levels and raised HDL-C level effectively occurred after the administration of OP for 8 weeks, which is supportive of a previous report published by S. J. Fan et al”
58. Page 11, line 238, “Insulin resistance (IR)…” changed to “IR…”
59. Page 11, line 238, “which is the…” changed to “which is another…”
60. Page 11, line 239, “polysaccharide” changed to “polysaccharides”
61. Page 11, line 240, “rodant anmial” changed to “rodent animal”
62. Page 11, line 240, “OP could be…” changed to “OP could…”
63. Page 11, line 242, “insulin resistance. IR can also… ” changed to “IR, which can also…”
64. Page 11, line 243, “PI3K/AKT signal pathway was widely…” changed to “Apparently, PI3K/AKT signal pathway is widely…”
65. Page 11, line 244, “pphosphorylation” changed to “phosphorylation”
66. Page 11, line 245, “cascade” changed to “cascades”
67. Page 11, line 245, “and subsequently…” changed to “that subsequently…”
68. Page 11, line 245, “the activated AKT phosphorylates GSK3β” changed to “phosphorylates GSK3β”
69. Page 11, line 246, “main factor that in charge of…” changed to “main factor in charge of…”
70. Page 11, line 247, “Similar to this study, our investigation also exhited that HFD feeding mice obviously increased the dephosphorylation of PI3K, AKT, and GSK3β in the liver. Interestingly, administration of OP (400 mg/kg) significantly increased the liver protein phosphorylation level of PI3K, AKT, and GSK3β…” changed to “Similarly, our investigation findings also exhibit that the dephosphorylation of PI3K, AKT, and GSK3β in the liver of HFD feeding mice obviously increased. In particular, the liver protein phosphorylation levels of PI3K, AKT, and GSK3β after the administration of OP (400 mg/kg) have significantly risen…”
71. Page 11, line 251, “which is also play a key…” changed to “which plays a key…”
72. Page 11, line 251, “oxidative stress. Accumulated…” changed to “oxidative stress, accumulated…”
73. Page 11, line 253, “Long term exposure of organs to hyperglycemic conditions, the generated free radicals will increase lead to…” changed to “The organs are exposed in long term hyperglycemic conditions, as a result, the generated free radicals will increase, leading to… ”
74. Page 11, line 254, “The defense system of antioxidant enzymes could be decrease the ROS to attenuate oxidative stress” changed to “In fact, the defense system of antioxidant enzymes may attenuate oxidative stress by reducing ROS”
75. Page 11, line 257, “oxidative stress. MDA is a biomarker for lipid peroxidation induced by oxidative stress” changed to “oxidative stress, which may induce lipid peroxidation with MDA as a biomarker”
76. Page 11, line 258, “The reduction of those antioxidant enzymes activities and elevated MDA level was observed…” changed to “It was observed that the activity of antioxidant enzyme decreased while MDA level increased”
77. Page 11, line 259, “their” changed to “previous”
78. Page 11, line 260, “we also shown that…” changed to “our study found that…”
79. Page 11, line 261, “could be…” changed to “could…”
80. Page 11, line 262, “and CAT and decrease the level of MDA…” changed to “and CAT while lower the level of MDA…”
81. Page 11, line 263, “clarified that…” changed to “proposed…”
82. Page 11, line 252, “these… ” changed to “the”
83. Page 11, line 265, “Nrf2 is an…” changed to “In addition, Nrf2 is an…”
84. Page 11, line 267, “in nuclear nucleus to protect redox homeostasis from oxidative injury” changed to “in nuclear nucleus, which may keep redox homeostasis from oxidative damage”
85. Page 11, line 267, “could be…” changed to “could…”
86. Page 11, line 268, “up-regulation the expression of HO-1 and SOD-2” changed to “of HO-1 and SOD-2 expression”
87. Page 11, line 269, “In our work, T2DM model mice reduced Nrf2 expression in the nucleus of the liver, indicating that T2DM model weakened defenses of…” changed to “In our work, T2DM model mice showed a reduced Nrf2 expression in liver nucleus, indicating that T2DM model weakened the defenses of …”
88. Page 12, line 270, “Contrarily, continuous…” changed to “Contrarily, the continuous…”
89. Page 12, line 272, “which was consistent with previous report” changed to “which is consistent with previous reports”
90. Page 12, line 273, “Mitochondrial dysfunction is another recognized as a prominent risk for T2DM” changed to “Additionally, mitochondrial dysfunction is recognized as another prominent risk of T2DM”
91. Page 12, line 274, “as one of major biomarker…which is associated with the…” changed to “as one of the major biomarkers… is associated with the…”
92. Page 12, line 275, “In our this study, we also showed that HFD feeding and STZ could be induce the activation of NOX2, while administration of OP (400 mg/kg) lead to induce the inactivation of NOX2, suggesting that inhidit NOX2 activation might be a new therapeutic strategy for the OP” changed to “In our study, it is also demonstrated that the activation of NOX2 could be induced by HFD feeding and STZ, while conversely the inactivation of NOX2 could be induced by the administration of OP (400 mg/kg), indicating that inhibition of NOX2 activation might be a new OP therapeutic strategy.”
93. Page 12, line 279, “supported” changed to “supports”
94. Page 12, line 279, “demonstrated” changed to “demonstrates”
95. Page 12, line 280, “and there was a possibility that OP could suppress NOX2 expression and enhance the Nrf2 level in the nucleus by activation of PI3K/AKT/GSK3β signal pathway in liver, because overproduction of ROS and MDA partly decreased, as well as SOD, CAT, GSH-Px increased after supplement with OP” changed to “Moreover, there is a possibility that OP may suppress NOX2 expression and enhance the Nrf2 level in the nucleus through activation of PI3K/AKT/GSK3β signal pathway in liver, because according to evidence, overproduction of ROS and MDA partly decreased, while SOD, CAT, GSH-Px increased after supplement with OP”
96. Page 12, line 283, “Intervention of okra as a functional food may be a new therapeutic strategy for alleviation oxidative stress and T2DM risks” changed to “As a functional food, okra introduced into intervention may be a new therapeutic strategy for alleviation of oxidative stress and reduction of T2DM risks”
97. Page 14, line 367, “The slice were…” changed to “The slice was…”
References
1. Slocum, S.L.; Skoko, J.J.; Wakabayashi, N.; Aja, S.; Yamamoto, M.; Kensler, T.W.; Chartoumpekis, D.V. Keap1/Nrf2 pathway activation leads to a repressed hepatic gluconeogenic and lipogenic program in mice on a high-fat diet. Arch. Biochem. Biophys., 2016, 591, 57-65.
2. Elango, B.; Dornadula, S.; Paulmurugan, R.; Ramkumar, K.M. Pterostilbene ameliorates streptozotocin-induced diabetes through enhancing antioxidant signaling pathways mediated by Nrf2. Chem. Res. Toxicol., 2016, 29, 47-57.
3. Zhang, J.X.; Wang, X.L.; Vikash, V.; Ye, Q.; Wu, D.D.; Liu, Y.L.; Dong, W.G. ROS and ROS-mediated cellular signaling. Oxid. Med. Cell. Longev., 2016. DOI: 10.1155/2016/4350965.

Round 2
Reviewer 2 Report
The current version of the paper is revised according to the comments. I think it can be considered for publication in Molecules.